# Green oxidation of indoles using halide catalysis

Jun Xu[1,2,3,4], Lixin Liang[2,4], Haohao Zheng[1], Yonggui Robin Chi[3] & Rongbiao Tong [2]*

Oxidation of indoles is a fundamental organic transformation to deliver a variety of synthetically and pharmaceutically valuable nitrogen-containing compounds. Prior methods require the use of either organic oxidants (*meta*-chloroperoxybenzoic acid, N-bromosuccinimide, *t*-BuOCl) or stoichiometric toxic transition metals [Pb(OAc)$_4$, OsO$_4$, CrO$_3$], which produced oxidant-derived by-products that are harmful to human health, pollute the environment and entail immediate purification. A general catalysis protocol using safer oxidants (H$_2$O$_2$, oxone, O$_2$) is highly desirable. Herein, we report a unified, efficient halide catalysis for three oxidation reactions of indoles using oxone as the terminal oxidant, namely oxidative rearrangement of tetrahydro-β-carbolines, indole oxidation to 2-oxindoles, and Witkop oxidation. This halide catalysis protocol represents a general, green oxidation method and is expected to be used widely due to several advantageous aspects including waste prevention, less hazardous chemical synthesis, and sustainable halide catalysis.

[1] College of Pharmacy, Guizhou University of Traditional Chinese Medicine, Guiyang, China. [2] Department of Chemistry, The Hong Kong University of Science and Technology(HKUST), Clear Water Bay, Kowloon, Hong Kong. [3] Division of Chemistry and Biological Chemistry, School of Physical and Mathematical Sciences, Nanyang Technological University(NTU), Singapore 637371, Singapore. [4] These authors contributed equally: Jun Xu, Lixin Liang. *email: rtong@ust.hk

Chemical oxidation of indoles is a fundamental organic transformation to deliver a diverse array of versatile nitrogen-containing compounds, in particular 2-oxindoles, which have been used widely in organic synthesis and drug discovery[1–4]. The electron-rich property of indoles allows the oxidation to occur under many oxidation conditions. However, a mixture of oxidation products is usually observed due to the competing oxidation of nitrogen, C2 and C3, as well as potential rearrangement and over-oxidation (Fig. 1a)[5–7]. The challenging chemo-selectivity and regio-selectivity requires not only a site-selective oxidant but also suitable substitutions at C2 and/or C3, as well as the protecting group on the nitrogen. Therefore, it is not surprising that only a small number of oxidants have been identified for only one or two of the three major types of the indole oxidation (Fig. 1a): (i) oxidative rearrangement of tetrahydro-β-carbolines to spirooxindoles[8–14], (ii) oxidation of C3-substituted indoles to 2-oxindoles[15], and (iii) oxidative cleavage of C2,C3-disubstituted indoles to 2-keto acetanilides (Witkop oxidation)[16,17]. Although these oxidants under the optimized conditions could solve the chemo-selectivity and regio-selectivity with high yields, their environmental and/or health impacts were not addressed, which is contrary to the rising concept and awareness of Green Chemistry.

Oxone (KHSO5-1/2KHSO4-1/2K2SO4, MW 307) has been widely used as a green, cheap, and safe oxidant because it generates $K_2SO_4$ as the only oxidant-derived byproduct[18]. Though inferior to oxygen ($O_2$) and hydrogen peroxide ($H_2O_2$) in terms of atom economy[19], oxone has not only admirable bench-stability for storage and transportation but also exceptional reactivity towards halide (e.g., bromide and chloride) oxidation under weakly acidic/ basic or even neutral conditions which is advantageous over related halide oxidation with hydrogen peroxide or oxygen under either strong acid (HCl or HBr) or transition metal catalysis[20]. We previously exploited its unique reactivity towards halide oxidation and have established several mild green oxone-halide protocols to replace the corresponding NXS-mediated oxidations[21,22]. For example, we found that oxone-halide could be used to replace NBS (or NCS) for oxidative halocyclization of tryptamine and tryptophol derivative (Fig. 1b)[22]. Inspired primarily by this work, we envision that in the absence of a tethered nucleophile the

indolenine (II) can react with water (part of the solvent) to generate 3-halo-2-hydroxy indoline (II → III, Fig. 1c), which might undergo semi-pinacol rearrangement[23] to provide spirooxindoles or 2-oxindoles. Alternatively, addition of potassium peroxymonosulfate (from excess of oxone) to indolenine (II) may generate hydroperoxysulfate intermediate IV (III → IV)[16,17]. Subsequent substitution of the halide with water triggers the C2-C3 bond cleavage of V to afford 2-keto acetanilides. In both scenarios, the halide is released and can be re-oxidized by oxone to generate the halogenating species. Therefore, halide is theoretically a catalyst for the oxone oxidation of indoles. This article presents the verification and implementation of this hypothesis, leading to the development of a unified green protocol for the oxidation of indoles to spirooxindoles, 2-oxindoles and 2-keto acetanilides (Fig. 1c). Our protocol (oxone-halide) can eliminate not only the use of hazardous oxidants (e.g., Pb(OAc)4, CrO3, OsO4, t-BuOCl, NBS, and m-CPBA, etc) but also the production of organic byproducts or toxic heavy metals derived from oxidants to minimize the environmental and health impact of the indole oxidation.

## Results

**Oxidative rearrangement of tetrahydro-β-carbolines.** The tricyclic spirooxindole core, in particular the spiro[pyrrolidine-3,3'-oxindole], is a privileged scaffold featured in a variety of medicinal agents (anti-tumor, anti-microbial, anti-viral, and anti-malarial, etc) and bioactive natural alkaloids (eg., spirotryprostatins, rhynchophylline, alstonisine, horsfiline)[24,25] (Fig. 2a). Oxidative rearrangement of tetrahydro-β-carbolines (THCs) to spirooxindoles was proposed as a biosynthetic process to account for the production of these metabolites and biogenesis connection with THC-type corynanthe alkaloids[26–28]. It has been successfully mimicked as a key transformation in the total synthesis of many spirooxindole alkaloids and thus oxidative rearrangement becomes a major approach for the synthesis of spirooxindoles[29]. However, the four identified stoichiometric oxidants: Pb(OAc)4[12], OsO4[30], t-BuOCl[8,9,11,12], and N-bromosuccinimide (NBS)[31] for the oxidative rearrangement are either unsafe to use or environmentally unfriendly. Pb(OAc)4 and OsO4 are extremely toxic heavy metal-based oxidants that pose a

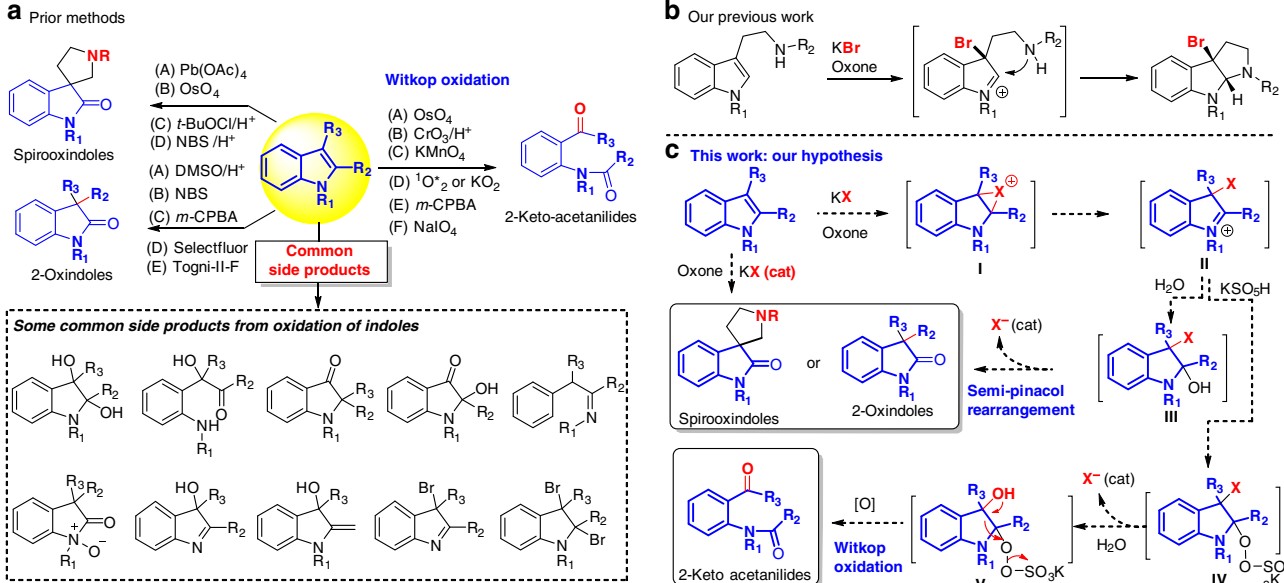

**Fig. 1** Oxidation of indoles and our hypothesis. **a** Prior methods for oxidation of indoles and some common side products; (**b**) Our previous work. **c** Hypothesis of oxone-halide oxidation of indoles. NBS N-Bromosuccinimide; m-CPBA meta-Chloroperoxybenzoic acid

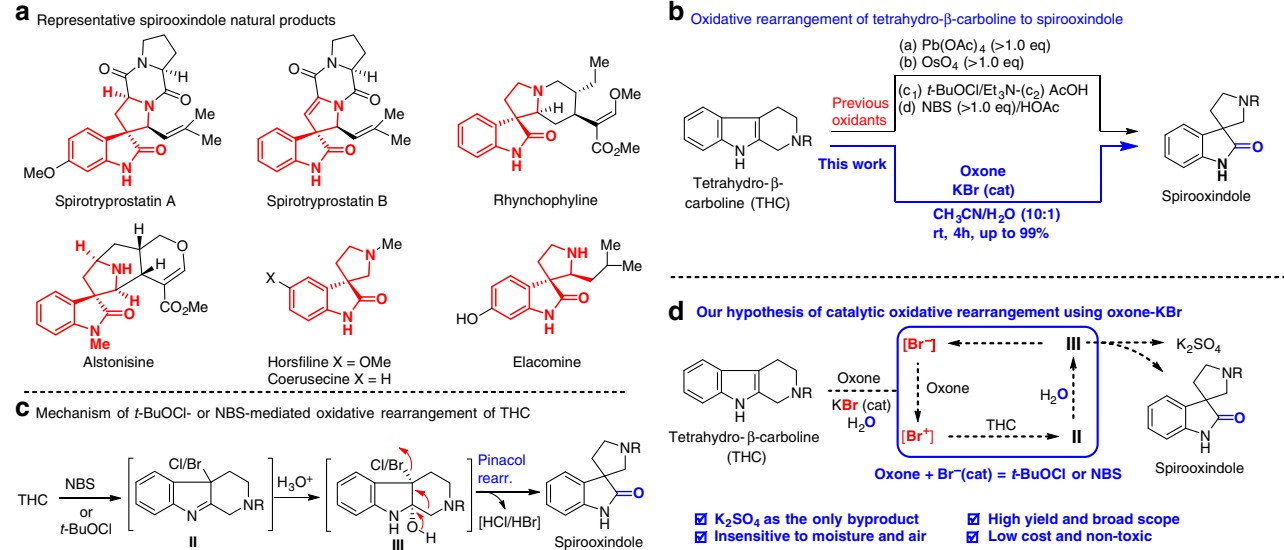

**Fig. 2** Spirooxindole natural products and oxidative rearrangement of tetrahydro-β-carbolines. **a** Representative spirooxindole natural products. **b** Oxidative rearrangement of tetrahydro-β-carboline to spirooxindole. **c** Mechanism of *t*-BuOCl or NBS-mediated oxidative rearrangement of tetrahydro-β-carbolines. **d** Our mechanistic hypothesis of oxidative rearrangement of tetrahydro-β-carboline to spirooxindole

significant threat to the human health and environment; while *t*-BuOCl is an unstable, flammable, harmful liquid that usually requires fresh in-house preparation and appropriate titration[32]. In addition, *t*-BuOCl required a subsequent acid treatment to complete the rearrangement (**II** → **III**, Fig. 2c). NBS was usually used in an acidic condition (AcOH–H$_2$O) and inevitably produced the corresponding stoichiometric succinimide byproduct that required immediate elimination by column purification. Therefore, a green catalytic protocol is highly desirable.

In continuation of our interest in developing green oxone-halide protocols to replace N-halosuccinimides (NXS) and related halogenating reagents (e.g., Cl$_2$, Br$_2$, *t*-BuOCl, etc), we set out to explore the oxidative rearrangement of THCs using oxone-halide as an green alternative (Fig. 2b) to the widely used NBS and *t*-BuOCl conditions. We believed such alternative was highly viable from the mechanistic perspective (Fig. 2c, d). The oxidative rearrangement of THCs involves a three-step sequence: oxidative halogenation, addition of water, and semi-pinacol rearrangement (Fig. 2c). We envisioned that oxone-halide (e.g., bromide) could deliver the reactive halogenating agent for the first step: oxidative halogenation. Small amount of water necessary for dissolving oxone as a co-solvent might add to β-bromo indolenine (**II**); while the halide released in the semi-pinacol rearrangement of **III** could be re-oxidized by oxone to re-generate the halogenating agent for the next-cycle THC oxidation (Fig. 2d). In principle, catalytic amount of halide (e.g., KBr) in combination of stoichiometric oxone could be used for replacement of *t*-BuOCl and NBS to achieve the goal of a green chemistry approach for the oxidative rearrangement of THCs to spirooxindoles.

To verify our hypothesis, we used THC **1a** as our model compound to examine its oxidative rearrangement under various conditions (Table 1). We quickly found that the combination of oxone (1.2 eq) and KBr (5 mol%) in both THF/H$_2$O (v/v = 1:1, 3:1, or 10:1) and MeCN/H$_2$O (v/v = 1:1, 3:1, or 10:1) effected the oxidative rearrangement within 4 h in excellent yields (84-93%) (Table 1, entries 1 and 2). As compared to NBS-AcOH (83% yield) and *t*-BuOCl-AcOH (79% yield), our protocol under optimal condition was higher yielding (93%). Other halides including tetrabutyl ammonium bromide (TBAB) (Table 1, entry 3), tetrabutyl ammonium

iodide (TBAI) (Table 1, entry 4), tetrabutyl ammonium chloride (TBAC) (Table 1, entry 5), KI (Table 1, entry 6), KCl (Table 1, entry 7), NH$_4$Cl (Table 1, entry 8) and NaCl (Table 1, entry 9), were also evaluated as the halide catalyst. We found that only TBAB was a competent halide catalyst without added advantage in terms of reaction time and yield. In the absence of halide (Table 1, entry 10), no rearranged product was observed in 24 h, which suggested that halide was the active catalyst for the oxidative rearrangement. In addition, other terminal oxidants including H$_2$O$_2$, K$_2$S$_2$O$_8$, NaOCl, NaClO$_2$, and *t*-BuOOH were examined but they were inferior to oxone (entries 11–15) because they were either unable to oxidize bromide (H$_2$O$_2$ and *t*-BuOOH, Table 1, entries 11 and 15) or unselective for oxidation of bromide and indole (K$_2$S$_2$O$_8$, NaOCl, and NaClO$_2$, Table 1, entries 12–14).

Next, we set out to examine the substrate scope (Table 2). It should be noted that, to the best of our knowledge, the substrate scope of this biomimetic oxidative rearrangement has not been studied systematically despite the fact that it was often used in the biomimetic total synthesis of spirooxindole alkaloids[29]. We first investigated the electronic effect of the protecting group (N–R$_1$ and N–R$_3$) on the nitrogen (Table 2, entries **2a**–**2j**). It was found that electron-donating group (EDG) including hydrogen, alkyl, and benzyl on the indole nitrogen (R$_1$ = H, alkyl, Bn) was essential to the success of oxidative rearrangement (Table 2, entries **2a**–**2c**). Interestingly, electron-withdrawing group (EWG, e.g., Ac, Ts, and Boc) on the indole nitrogen (N–R$_1$) resulted in lower conversion (20–50%) and loss of EWG (**2a** was obtained instead of the expected **2d**). The high chemoselectivity of indole oxidation via halide catalysis is hinged on that in situ generated halenium ion (c.f., Br+) as a catalyst reacts only with electron-rich indole (C2=C3) to form the corresponding indole halonium intermediate (**I**, Fig. 1c).

An electron-withdrawing group on the indole nitrogen (R$_1$ = Ts, Boc) will substantially decrease the electron-density of indoles and consequently suppress the halenium-catalyzed indole oxidation, which is consistent with the result of **2d** with 0% yield (R$_1$ = Ts or Boc) in Table 2. On the other hand, the electronic property of protecting group on the piperidine (N-R$_3$) is less significant to the electron density of indoles (not a conjugate system) and less influential to their oxidation under the halide catalysis, which is

**Table 1 Selected conditions for oxidative rearrangement of THC 1a**

| Entry | Oxidant (1.2 eq) | MX (5 mol%) | Solvents (v/v, 10/1) | Conv. (%) | Yield (%)$^a$ |
|---|---|---|---|---|---|
| 1 | Oxone | KBr | THF/$H_2O$ (1/1→10/1) | 100 | 87–93 |
| 2 | Oxone | KBr | MeCN/$H_2O$ (1/1→10/1) | 100 | 84–93 |
| 3 | Oxone | TBAB | MeCN/$H_2O$ | 100 | 89 |
| 4 | Oxone | TBAI | MeCN/$H_2O$ | 55 | 33 |
| 5 | Oxone | TBAC | MeCN/$H_2O$ | 40 | 19 |
| 6 | Oxone | KI | MeCN/$H_2O$ | 64 | 37 |
| 7 | Oxone | KCl | MeCN/$H_2O$ | 28 | 19 |
| 8 | Oxone | $NH_4Cl$ | MeCN/$H_2O$ | 30 | 17 |
| 9 | Oxone | NaCl | MeCN/$H_2O$ | 32 | 16 |
| 10 | Oxone | – | MeCN/$H_2O$ | 20 | 0 |
| 11 | $H_2O_2$ | KBr | MeCN/$H_2O$ | <10 | 0 |
| 12 | $K_2S_2O_8$ | KBr | MeCN/$H_2O$ | 25 | 12 |
| 13 | NaClO | KBr | MeCN/$H_2O$ | 75 | 54 |
| 14 | $NaClO_2$ | KBr | MeCN/$H_2O$ | <10 | 0 |
| 15 | t-BuOOH | KBr | MeCN/$H_2O$ | <10 | 0 |

$^a$yield was obtained by $^1$H-NMR analysis of the crude product using $CH_2Br_2$ as the internal reference. TBAB: tetrabutylammonium bromide; TBAC: tetrabutylammonium chloride; TBAI: tetrabutylammonium iodide

consistent with the higher yield (90–99%) with N-EWG (Table 2, entries **2e**–**2h**: N-Boc, N-Ts, N-Cbz, and N-Ac). The low conversion (10–20%) for EDG (Table 2, entries **2i**–**2j**: N–Me and N–Bn) under neutral condition might be attributed to preferentially oxidize the tertiary amine. It was later found that an acidic medium (THF/AcOH/$H_2O$ = 1:1:1) could substantially improve the conversion (>90%) and yield (60-74%). The lower isolated yield for electron-donating N-$R_3$ (Table 2, entries **2i**, **2j**, **2w**, **2x**, and **2y**) products might be due to the difficult isolation/purification of tertiary amine products from our acidic reaction medium. Two examples with electron-donating substituents (Me and OMe) on the THCs were selected to probe the possibly competing bromination on the benzene ring of THCs under our halogenating condition. Fortunately, the oxidative rearrangement occurred smoothly under our optimized condition (Table 2, entries **2k** and **2l**) and no aromatic bromination was detected. This finding was important to relevant total synthesis because such EDG substituents are found in a number of natural products (spirotryprostatin A, horsfiline and elacomine). Next, a variety of C1-substituted THCs were examined for their oxidative rearrangement (Table 2, entries **2m**–**2y**). Except for C1-aryl THC (Table 2, entry **2v**, 0%) that resulted in unexpected oxidative C1–N3 cleavage (see Supplementary Figs. 92 and 93 for the structure of this side product and possible mechanisms), all C1-substituted THCs with various functional groups (alkene, CN, OBn, alkyne, and $CO_2Et$) underwent smooth oxidative rearrangement to give the spirooxindole products in good to excellent yields with diastereoselectivity ranging from 1.5:1 to 3.8:1. Most of these diastereomers could be separated easily by column chromatography on silica gel and their relative stereochemistry was proposed according to the relative configuration of **2o** (3R*/4S*) and **2o'** (3R*/4R*), which were confirmed by X-ray diffraction analysis. It was to our surprise that **2u** was obtained in 80% yield as a single diastereomer (dr 20:1, 3R*/4S*). This remarkable high diastereoselectivity was in sharp contrast to those of tryptophan-derived THC **2z** (dr 4:1). Interestingly, we found that C3-ester substitution enhanced the stereocontrol of C1-alkyl on the spirocenter from 1.5/1–3.8/1 (Table 2, entries **2m/m'**–**2t/t'**) to 7/1–20/1 (Table 2, entries **2ac** and **2ab**). Another

unexpected observation was that electron-donating group on the piperidine nitrogen ($R_3$) appeared to reverse such diastereoselectivity, leading to isolation of the major products (Table 2, entries **2w** and **2y**) with different relative stereochemistry (3R*/4R*). The intriguing diastereoselectivity was not documented in the literature and our finding would be instrumental to the design and synthesis of spirooxindoles from THCs.

To showcase the utility of this protocol (Fig. 3), we achieved the total synthesis of two popularly targeted spirooxindole natural products (±)-coerulescine (1.2 g, **2i**) and (±)-horsfiline (**3**) from THC **1a** (Fig. 3a)[31,33–37]. Reduction of THC **1a** with LiAlH₄ and oxidative rearrangement of the resulting THC **1i** using our oxone-KBr under acidic condition (THF/AcOH/$H_2O$ = 1:1:1) furnished (±)-coerulescine (**2i**) with 1.2 g in two steps (overall yield: 39%). If the oxidative rearrangement of THC **1i** was carried out with stoichiometric KBr and 2.4 equivalent of oxone, sequential one-pot oxidative rearrangement and bromination occurred to provide C5-bromo spirooxindole **2ad** in 41% yield, which could be used for CuI-catalyzed Ullmann ether synthesis to furnish (±)-horsfiline (**3**) in 60% yield. Notably, this protocol allowed a one-pot sequential oxidative rearrangement and dibromination (**1a** → **2ae**, 86% yield) when 2.1 equivalent of KBr and 3.6 equivalent of oxone were employed. This offered a compelling flexibility to access to a wide variety of spirooxindoles. Finally, we applied this protocol for the biomimetic oxidative rearrangement of natural alkaloid yohimbine and obtained the corresponding yohimbine oxindole **4** (Fig. 3b) in 56% yield, which apparently was superior to the reported three-step method[38] with only 38% overall yield.

To further extend this protocol, we were interested in the rarely-explored oxidative rearrangement of 1,3,4,9-tetrahydropyrano[3,4-b]indoles[39,40] (THPIs, **5a**–**5e**) to the oxa-spirooxindoles (Fig. 3d) because (i) oxa-spirooxindole is the structural core in many pharmaceutically important molecules[41,42] (Fig. 3c) and (ii) there are only a few synthetic methods[43,44]. To our delight, without further condition optimization all five THPIs underwent the expected oxidative rearrangement to provide the oxa-spirooxindoles **6a**–**6d** in good to excellent yields, which constitutes the second examples of oxidative rearrangement of

**Table 2 Substrate scope of oxidative rearrangement of tetrahydro-β-carbolines**

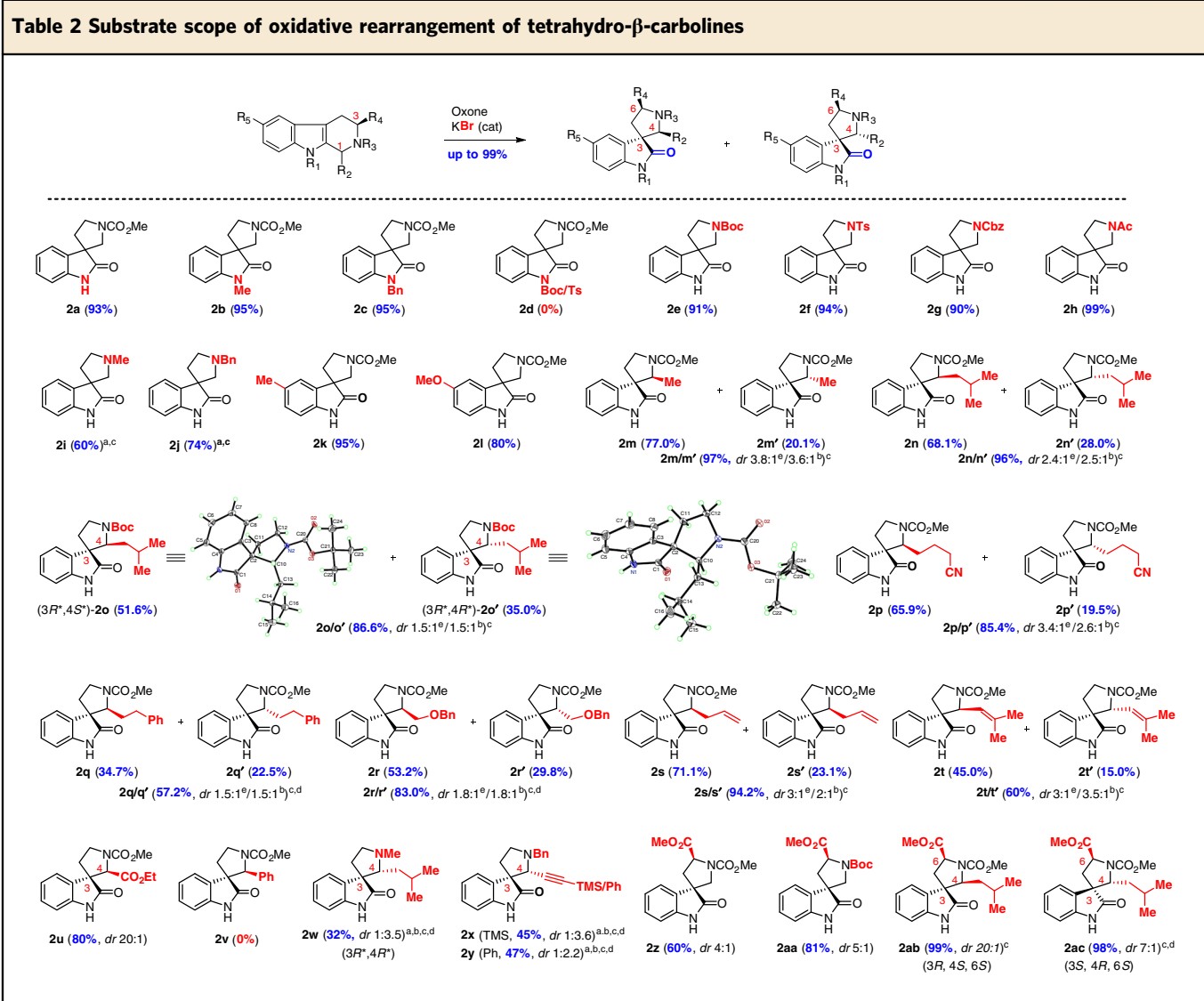

<sup>a</sup>The reaction was carried out in THF/AcOH/H2O (1:1:1) at room temperature for 0.5–20 h
<sup>b</sup> The minor diastereomer could not be obtained and the diastereomeric ratio was determined by 1H-NMR analysis of the crude reaction mixture
<sup>c</sup>10 mol% KBr was used
<sup>d</sup>Additional 0.6 equivalent of oxone was added after 12 h reaction
<sup>e</sup>Isolated diastereomeric ratio

THPIs[45]. Remarkably, the diastereoselectivity (3R*/4R*) was unusually high and only a single diastereomer was isolated. The relative stereochemistry of **6a**, **6c**, and **6e** was confirmed by X-ray diffraction. Notably, oxa-spirooxindole **6a** possessed the same relative configuration as coixspirolactam C[46,47] and could be used as a direct precursor for the synthesis of coixspirolactam C[45].

**Oxidation of indoles to 2-oxindoles**. The success of the green approach for the oxidative rearrangement of THCs/THPIs to (oxa-)spirooxindoles prompted us to explore the possibility of the oxone-halide oxidation of the simpler C3-substituted indoles to 2-oxindoles. 2-Cxindoles are not only important structural motifs in a number of biologically active alkaloid natural products and pharmaceutical molecules[47] but also frequently used as the synthetic building blocks in the synthesis of natural alkaloids and as the platform for development of synthetic methodologies[48]. As shown in Fig. 4, the prior methods for direct oxidation of indoles to 2-oxindoles employed usually NBS[5] or m-CPBA[49] as the

stoichiometric oxidant, even though electrophilic fluorinating agents such as selectfluor[50] and Togni's reagent[15] were found to be ideal oxidants for some specific indoles that suffered from low yields when using NBS and m-CPBA. The DMSO-HCl (37%) condition[51] was often limited to the oxidation of simple indoles without acid-labile functional groups. Apparently, there lacks of a green and efficient method for the indole oxidation to 2-oxindoles. We believed that the green oxone-halide oxidation system could be applicable to this case.

We chose 3-methylindole (skatole, **7a**) as our model compound to examine the direct oxidation of indoles to 2-oxindoles. After quick screening of various solvents (Table 3, entries 1–9), three solvent systems: THF/H<sub>2</sub>O (20:1), MeCN/H<sub>2</sub>O (20:1), and t-BuOH/H<sub>2</sub>O (20:1), were identified to be an excellent reaction medium. We selected t-BuOH/H<sub>2</sub>O (20:1) for the best yield (91%) of 3-methyloxindole (**8a**) from the skatole oxidation. Notably, KBr was essential (Table 3, entry 13) and outperformed the corresponding KCl (58%) and KI (0%) (Table 3, entries 10 and 11), while the higher water ratio in the mixed solvent system

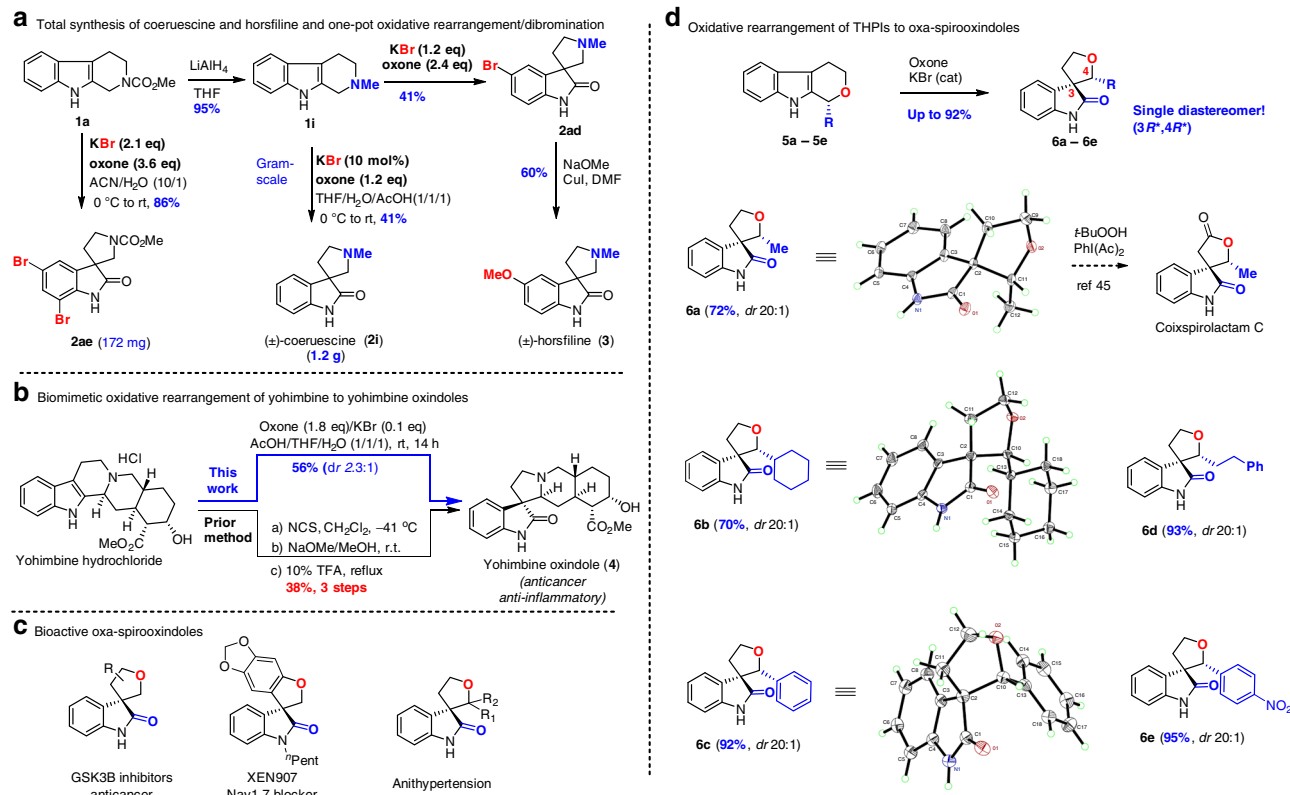

**Fig. 3** Synthetic utility of our halide catalysis for the oxidative rearrangement. **a** Total synthesis of (±)-coeruscine and (±)-horsfiline and one-pot bromination of spirooxindole. **b** Oxidative rearrangement of Yohimbine to β-Yohimbine oxindole. **c** Bioactive molecules bearing oxa-spirooxindole. **d** Oxone-KBr oxidative rearrangement of tetrahydropyrano[2,3-b]indoles (THPIs) to oxa-spirooxindoles

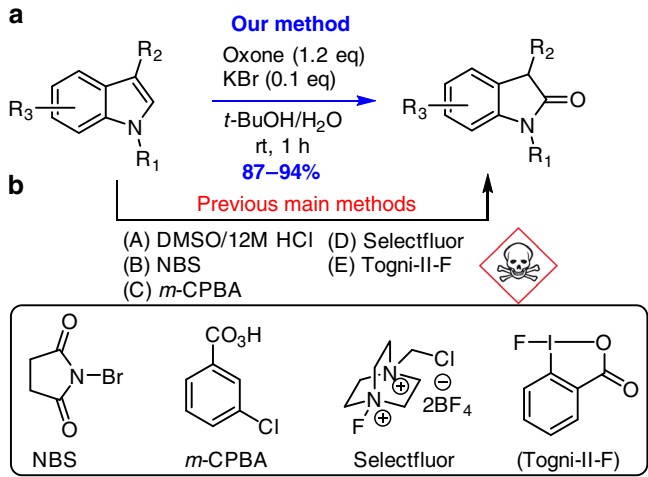

**Fig. 4** Oxidation of indoles to 2-oxindoles. **a** Our method for oxidation of indoles to 2-oxindoles. **b** Previous methods for oxidation of indoles to 2-oxindoles

only slightly lowered the yield of oxidation (Table 3, entry 12). It was noted that our oxone-KBr protocol was more efficient than other methods (Togni's reagent: 77%; NBS: 26–83%; CH₃CO₃H: 14%) for skatole oxidation.

This optimized condition was applied to oxidation of a variety of C3-substituted indoles (Fig. 5a). Our examination of the substrate scope led to some interesting findings. First of all, the electronic property of the protecting group on the indole nitrogen has a dominant influence on the oxidation: electron-donating groups including methyl, benzyl, allyl ($R_1$ = Me, Bn, Allyl) were

favorable (**8b**-**8c**, 90–92%), while electron-withdrawing group (e.g., $R_1$ = Ts or Boc) completely suppressed the oxidation (**8d'**, 0%). Secondly, the electronic properties of C3 substituents ($R_2$) is critical to the success of oxidation: C3-alkyl (electron-donating) indoles gave high yields while the parent indole, C3-phenylindole and C3-trifluoromethyl failed to deliver the corresponding 2-oxindoles (**8a'**, **8a''**, and **8a'''** 0%). This finding was consistent with the observation that indole oxidation via halide catalysis required electron-rich indoles. The parent indole suffered from poor chemo-selectivity and regio-selectivity and gave a complex mixture, which was observed by *m*-CPBA. Thirdly, substitution at C5 and C7 of indoles ($R_3$ = 7-Me, 5-Br, and 5-OMe) has little effect on the oxidation (**8e**-**8g**, 88–91%). Moreover, our oxone-KBr protocol was applicable to tryptamines (**8j**–**8p**), tryptophans (**8q**), tryptophols (**8w**–**8z**), and their derivatives (**8r**–**8v**). In addition, the esters (**8q**-**8s**), carbamate (**8j**, **8q**), sulfonamides (**8k**-**8l**, **8n**-**8p**), cyanide (**8t**-**8v**), and even free alcohol (**8w**-**8z**) were tolerated in this mild oxidation condition, which out-performed prior methods regarding the functional group tolerance and efficiency.

To showcase the scalability and utility of this oxone-KBr oxidation process (Fig. 5b), the catalytic oxidation of **7b** and **7h** on 2.0 mmol (2.62 g and 4.15 g, respectively) scales was carried out to provide the desire 2-oxindoles **8b** and **8h** in the excellent yield of 91% and 88%, respectively, which were used for the concise unified total syntheses of desoxyeseroline, physovenol methyl ether, and esermethole[48,52–54]. The availability of 2-oxindoles **8b** and **8h** with gram quantities enabled alkylation with two-carbon bromides[55] to provide the 3,3-disubstituted 2-oxindoles (**9a**–**9d**). Reductive cyclization[56] was employed for the construction of the key tricyclic hexahydropyrroloindolines (HPIs, **10c** and **10d**) and tetrahydrofuroindolines (TFIs, **10a** and

10b). N-Methylation of the resulting hexahydropyrroloindoline furnished desoxyeseroline (10c) in 70% yield (46% overall yield for three steps). CuI-catalyzed Ulmann coupling of aryl bromide with NaOMe completed the synthesis of physovenol methyl ether

(11a) and esermethole (11b) in 72% and 74%, respectively. Notably, physovenol methyl ether and esermethole were the 2-step precursor of respective physovenine and physostigmine[57].

In order to shed some light on the oxidation mechanism, we performed a small set of controlled experiments (Fig. 5c). 2-Deuterated 3-methyindole(D-7a, 72% D) was prepared and used for oxone-KBr oxidation. 49% Deuterium incorporation at C3 was observed in D-8a (90% yield). When $D_2O$ was used as a co-solvent, 85% deuterium at C3 was observed for the oxidation of un-deuterated substrate 7a. This seemingly contradictory result was attributed to the keto-enol tautomerism of 2-oxindole 8a under either neutral condition (THF/$D_2O$, 14%D) or our standard condition (21%D). C2-Bromoindole (7aa) was not the intermediate for our oxone-KBr oxidation because it failed under our condition to provide 2-oxindole 8a. In addition, no 2-oxindole 8a was observed from the oxidation of 2-methylindole (7ab), which suggested that C3-alkyl substitution stabilized the developing positive charge at C3 in the course of bromide departure. All these results supported the proposed mechanism (Fig. 1c) that involved semi-pinacol rearrangement to provide the 2-oxindoles. However, we could not exclude the possible H–Br elimination over semi-pinacol rearrangement to afford 2-oxindoles when C2 was unsubstituted.

**Witkop oxidation of indoles to 2-keto acetanilides**. Oxidative cleavage of aromatic rings occurs frequently in Nature[58]. In particular, the enzymatic oxidation of tryptophan to N-formylkynurenine is not only a major metabolic pathway of tryptophan but also the first key step of the biosynthesis of coenzyme NAD[59]. The first chemical process of the corresponding oxidative cleavage of the C2–C3 double bond of indoles was reported in 1951 by Witkop[16,17] using Pt/$O_2$ oxidation (Fig. 6b). Subsequently, various oxidants including peracids

**Table 3 Selected conditions for oxone-KBr oxidation of skatole[a]**

Skatole (7a) → 8a, KX, oxone, Solvent, rt

(a) Oxone-KBr: 91%
(b) Togni-II-F: 77%
(c) NBS: 26–83%
(d) CH₃CO₃H: 14%

| Entry | KX(10 mol%) | Solvents (v/v) | Time (h) | Yield (%)[b] |
|---|---|---|---|---|
| 1 | KBr | MeOH | 2 | <5 |
| 2 | KBr | tBuOH | 2 | <5 |
| 3 | KBr | $CH_2Cl_2$ | 2 | <5 |
| 4 | KBr | DMF | 2 | <5 |
| 5 | KBr | $H_2O$ | 2 | <5 |
| 6 | KBr | DMSO | 2 | <5 |
| 7 | KBr | THF/$H_2O$ (20:1) | 1 | 87 |
| 8 | KBr | MeCN/$H_2O$ (20:1) | 1 | 85 |
| 9 | KBr | tBuOH/$H_2O$ (20:1) | 1 | 91 |
| 10 | KCl | tBuOH/$H_2O$ (20:1) | 2 | 58 |
| 11 | KI | tBuOH/$H_2O$ (20:1) | 4 | 0 |
| 12 | KBr | tBuOH/$H_2O$ (10:1) | 1 | 87 |
| 13 | – | tBuOH/$H_2O$ (20:1) | 4 | 0 |

[a]The reaction was carried out at room temperature with skatole (0.5 mmol), oxone (0.6 mmol), KX (10 mol%), solvent (5.0 mL)
[b]Isolated yield was obtained

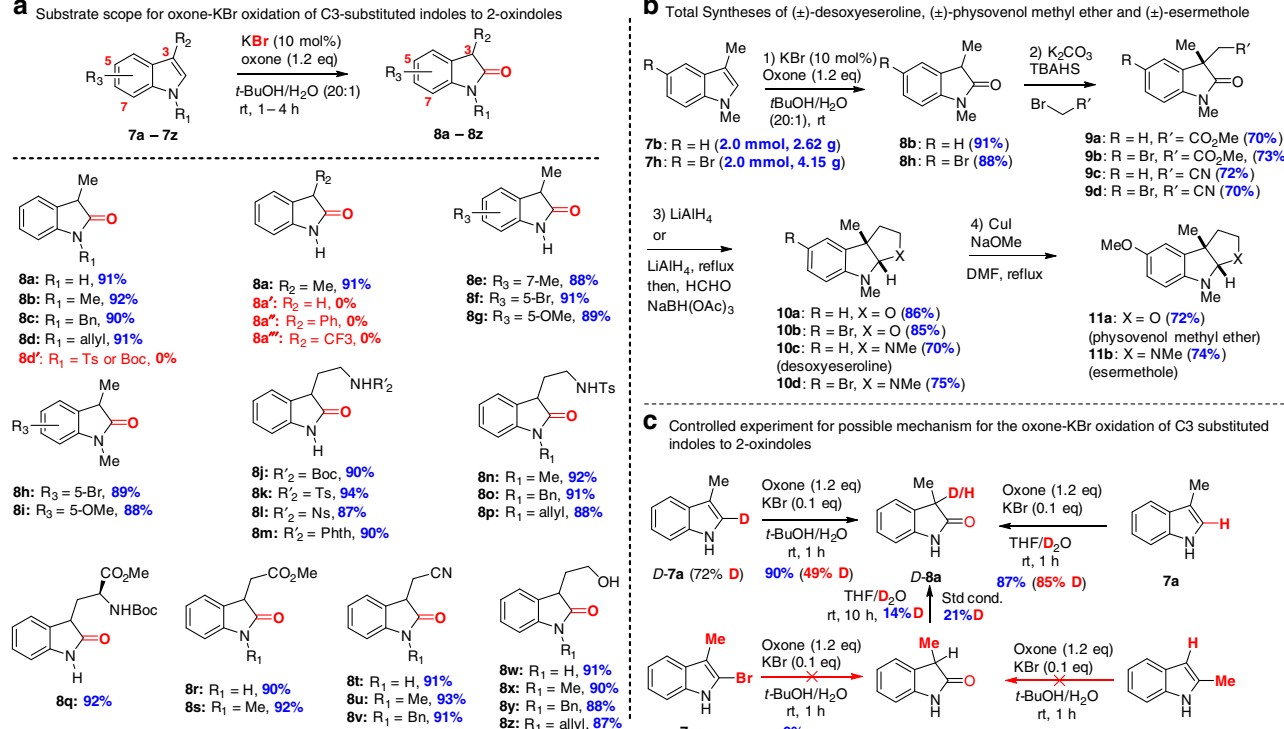

**Fig. 5** Oxone-Halide oxidation of indoles to 2-oxindoles. **a** Substrate scope for oxone-KBr oxidation of C3-substituted indoles to 2-oxindoles. **b** Total syntheses of (±)-desoxyeseroline, (±)-physovenol methyl ether and (±)-esermethole. **c** Controlled experiments for possible mechanism for the oxone-KBr oxidation of C3 substituted indoles to 2-oxindoles. TBAHS Tetrabutylammonium hydrogen sulfate

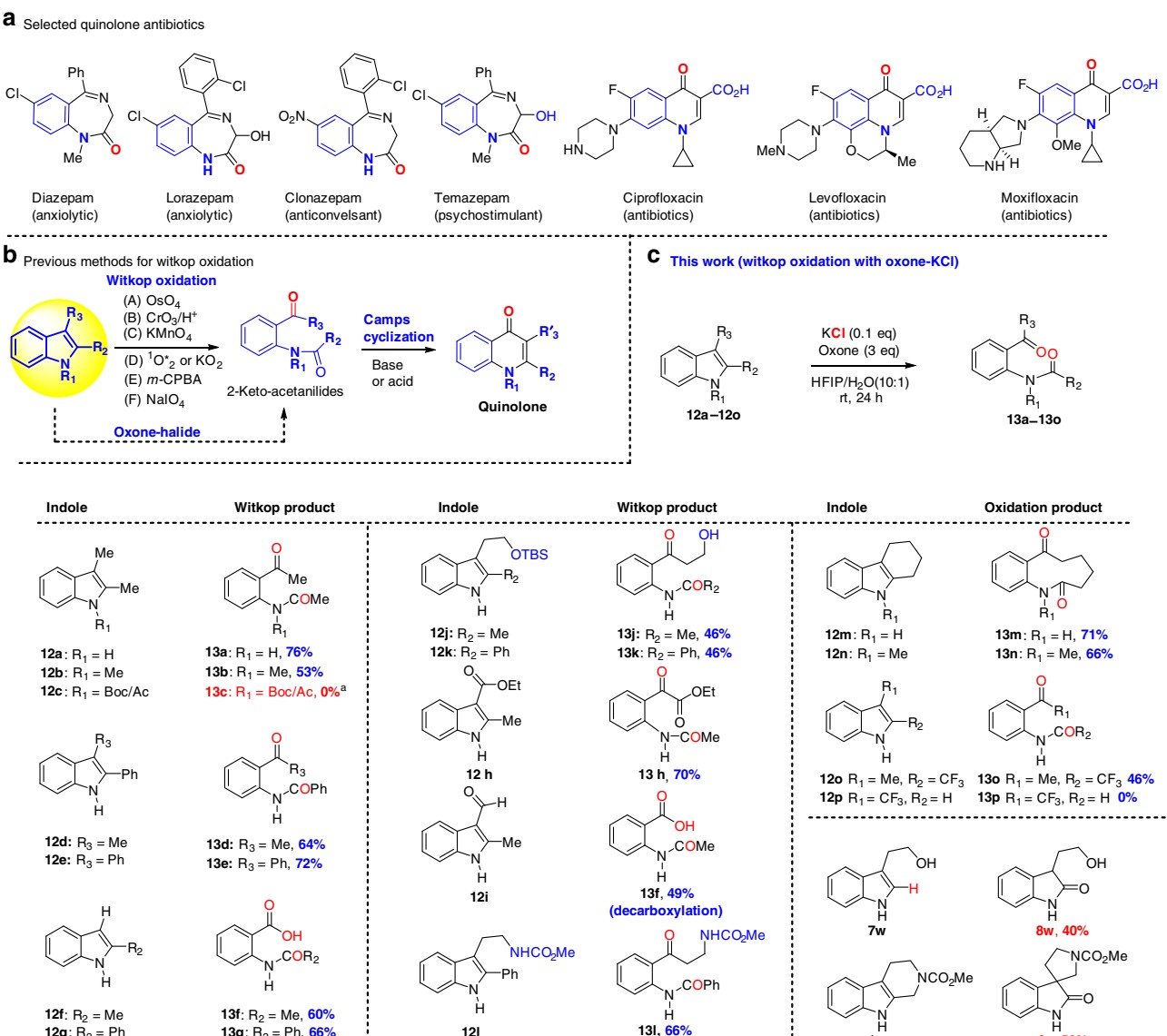

**Fig. 6** Quinolone antibiotics and Witkop oxidation. **a** Selected quinolone antibiotics. **b** Previous methods for Witkop oxidation. **c** Witkop Oxidation with Oxone-KCl (This work)

(*m*-CPBA), periodic acid (NaIO₄), chromic acid, ozone and singlet oxygen, were identified for Witkop oxidation[16,17,60–64]. Notably, Winterfeldt[65] found that NaH/O₂ and KO$^t$Bu/O₂ could effect both Witkop oxidation and Camps cyclization to provide quinolones, which widely exist in many marketed drugs and bioactive molecules[66] (Fig. 6a). The importance of Witkop oxidation of indoles to 2-keto-acetanilides for the Camps cyclization to quinolones and for the commercial preparation of benzodiazepines[67,68] (drugs for treatment of insomnia and anxiety) aroused our interest in developing a green oxidation protocol for Witkop oxidation using the oxone-halide system (Fig.6c).

We chose 2,3-dimethyl indole (**12a**) as the model compound to examine the viability of Witkop oxidation with the oxone-halide system. Surprisingly, the optimized conditions developed for oxone-halide oxidation of indoles to spirooxindoles and 2-oxindoles afforded only 14–34% yield of fragmentation product **13a** (Table 4, entries 1–3). A large-scale screening of solvents (Table 4, entries 4–6) and halides (Table 4, entries 7–9) enabled us to identify a clean and efficient system: oxone-KCl in HFIP/H₂O (10:1) (Table 4, entry 8), which could deliver the desired

Witkop product **13a** in 74%. It was noted that the reaction time should be extended to 24 h, which was much longer than the time required for the oxidation of indoles to spirooxindoles and 2-oxindoles (1–4 h).

The success of Witkop oxidation with oxone-halide system in our model study prompted us to investigate the substrate scope (Fig. 6c). It was found electron-withdrawing group on the indole nitrogen (N-Ac or N-Boc) could not allow for the Witkop oxidation with oxone-KCl. One of our major findings in the course of expanding substrate scope was that C2 substitution (R₂ ≠ H) was necessary for the oxidative cleavage (**12a–12o**) as the C2-unsubstituted indole **7w** only resulted in 2-oxindole **8w** (~40%). Although C2,C3-disubstituted indoles were excellent substrates for Witkop oxidation with oxone-halide in HFIP/H₂O system, C3-substitution was not critical (e.g., **12f** and **12g**: R₃ = H). In the latter case, it should be noted that the oxone-KCl oxidation led to isolation of carboxylic acids **13f** and **13g**, instead of the expected aldehydes. Another interesting observation was that oxone-KCl oxidation of C3-aldehydic indole **12i** provided the unexpected carboxylic acid **13f**, which might be arisen from an oxidation sequence involving C2–C3 cleavage, aldehyde

**Table 4 Selected conditions for Witkop oxidation with oxone-halide**

| Entry | KX (10 mol%) | Solvents (v/v) | Time (h) | Yield (%) |
|---|---|---|---|---|
| 1 | KBr | THF/$H_2O$ (10:1) | 24 | 20 |
| 2 | KBr | MeCN/$H_2O$ (10:1) | 24 | 34 |
| 3 | KBr | $t$BuOH/$H_2O$ (10:1) | 24 | 14 |
| 4 | KBr | acetone/$H_2O$ (10:1) | 24 | 34 |
| 5 | KBr | DMSO | 24 | <5 |
| 6 | KBr | HFIP | 24 | 47 |
| 7 | KBr | HIFP/$H_2O$ (10:1) | 24 | 69 |
| 8 | KCl | HIFP/$H_2O$ (10:1) | 24 | 74 |
| 9 | KI | HIFP/$H_2O$ (10:1) | 24 | 13 |

The reaction was carried out with 12a (0.1 mmol, 1.0 eq), oxone (0.3 mmol, 3.0 eq), KX (10 mol %), solv ent (1.0 mL, 0.1 M), room temperature. The yield was determined by 1H-NMR analysis of the crude reaction mixture
*HFIP* Hexaf luoroisopropanol

oxidation, and oxidative decarboxylation. We recognized that our oxone-KCl in HFIP/$H_2O$ was too acidic for *tert*-butyldimethylsilyl ethers (12i and 12k), leading to desilylated Witkop oxidation products (13j and 13k). Fortunately, the free alcohol survived from this oxidation condition. It was intriguing to observe that 1,2,3,4-tetrahydrocarbazoles (12m and 12n) could undergo smoothly oxidative C2–C3 cleavage, while the corresponding tetrahydro-β-carboline 1a only resulted in oxidative rearrangement under the identical condition (oxone-KCl in HFIP/$H_2O$). At this stage, without further experimentation we could not provide a good explanation for this puzzling result. Finally, we examined the Witkop oxidation of indoles with $CF_3$ substitution at C2/C3 (12o/12p) and found that C2–$CF_3$ indole could deliver the desired Witkop product 13o in 46% while 12p decomposed under the condition. Nevertheless, the property of substituents did play a decisive role in the oxidation of indoles to different products and we believed that our result would support the hypothetic mechanism of Witkop oxidation in Fig. 1c.

## Discussion

We have developed a general halide catalysis for green oxidation of indoles to spirooxindoles, 2-oxindoles, 2-keto acetanilides. Our study demonstrated that oxone-halide could replace other organic halogenating agents (NBS, NCS, *t*-BuOCl etc) or peracids (*m*-CPBA) in different types of oxidation of indoles, and thus eliminate the production of toxic organic byproducts derived from oxidants. As compared to prior methods, this protocol was usually more efficient partly due to the in situ generated halenium ion ($X^+$) catalyst that has the appropriate concentration and reactivity towards the C2–C3 double bond of indoles and thus significantly suppressed other competing oxidations/rearrangements. In addition, no need to protect the indole nitrogen was advantageous since most previous methods required to mask the indole nitrogen with electron-withdrawing groups (e.g., N-Ts, N-Boc, N-Ac etc) for better chemo-selectivity and regio-selectivity. Achieving this oxone-halide oxidation of indoles was a milestone in the indole oxidation for its low-cost, safe/simple operation (open flask), and most importantly its greenness in several aspects of the 12 Green Chemistry Principles including (1) preventing waste, (2) less hazardous chemical synthesis, (3) safer chemicals,

and (4) using catalysis. We believed that this oxone-halide system might be used for other types of indole oxidation that were not explored in this article. It is our expectation that this oxone-halide protocol for the indole oxidation will find wide applications in academia (organic synthesis) and industrial (pharmaceutical) communities.

## Methods

**Oxidative rearrangement of tetrahydro-β-carbolines**. To a stirred solution of THC (1.0 eq) and KBr (5–10 mol%) in MeCN/$H_2O$ (10/1, 0.1 M) or in THF/$H_2O$/AcOH(1/1/1, 0.1 M) at 0 °C was added oxone (1.2 eq, MW = 307) in one batch. The resulting solution was allowed to warm to rt, and stirred for 1–16 h. After the reaction was completed as determined by TLC analysis, the reaction was quenched by addition of *aq. sat.* NaHCO$_3$ and *aq. sat.* Na$_2$SO$_3$, and then diluted with EtOAc. The organic fractions were collected, and the aqueous phase was extracted with EtOAc three times. The combined organic fractions were washed with brine, dried over Na$_2$SO$_4$, filtered, and concentrated under reduced pressure. The resulting residue was purified by silica gel column chromatography to give spirooxindoles.

**Oxidation of C3-substituted indole to 2-oxindole**. To a solution of C3-substituted indole (1.0 eq) and KBr (10 mol%) in *t*-BuOH/$H_2O$ (20/1, 0.1 M) at rt was added oxone (1.2 eq, MW = 307), and was stirred for 1–4 h. The reaction was quenched by addition of *aq. sat.* Na$_2$SO$_3$ and then diluted with EtOAc. The organic fractions were collected and the aqueous phase was extracted with EtOAc three times. The combined organic franctions were washed with brine, dried over Na$_2$SO$_4$, filtered, and concentrated under reduced pressure. The resulting residue was purified by silica gel column chromatography to give the 2-oxindoles.

**Witkop oxidation of indole to 2-keto acetanilide**. To a solution of indole (12a–12n, 1.0 eq) and KCl (10 mol%, 0.01 M) in HFIP/$H_2O$ (10/1, 0.1 M) at rt was added oxone (1.2 eq, MW = 307) in one batch. The resulting solution was stirred at room temperature for 24-h and then diluted with EtOAc. The reaction mixture passed through a short pad of silica gel and washed with EtOAc. The resulting EtOAc/HFIP solution was concentrated under reduced pressure and the residue was purified by flash column chromatography to give 2-keto acetanilides.

## Data availability

Experimental procedures and characterization data are available within this article and its Supplementary Information. Data are also available from the corresponding author on request. The X-ray crystallographic coordinates for structures reported in this study have been deposited at the Cambridge Crystallographic Data Center (CCDC), under deposition numbers 1935503, 1935504, 1935506, 1935507, and 1935508. These data can be obtained free of charge from The Cambridge Crystallographic Data Center via www.ccdc.cam.ac.uk/data_request/cif.

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

## Acknowledgements

This research was financially supported by Research Grant Council of Hong Kong (16311716, 16303617, 16304618) and National Natural Science Foundation of China (21772167). Dr. J.X. also acknowledged the Doctor Start-up Fund ([2018]28) and the Guizhou Province First-Class Disciplines Project (Yiliu Xueke Jianshe Xiangmu-GNYL [2017]008) from Guizhou University of Traditional Chinese Medicine (China).

## Author contributions

J.X. and L.L. performed the experiments. H.Z. prepared some related substrates. Y.R.C. participated in discussing part of the experiments. R.T. conceptualized and directed the project, and drafted the paper with the assistance from all co-authors.

## Competing interests

The authors declare no competing interests.
