## [Peer Review File · Nature Communications]

Reviewers' comments:

Reviewer #1 (Remarks to the Author):

I think this is an important communication that introduces halide-type catalysis for controlled oxidation of indoles, with control over the possible rearrangement, or overoxidation pathways that stem from the activated indole intermediates. A nice case is made for the "green" nature of the new protocols, and this reviewer feels that this is valid.

The discussion of background and context is quite good, with a most welcome tribute to some of the pioneers of indole oxidation from the older literature. The authors should add the following references to the list as some similar goals are achieved with H₂O₂-based oxidations and biomimetic catalysts. Some of the uncited work is also relevant to the synthetically useful rearrangements noted in the present submission. So, these two citations really should be added.

J. Am. Chem. Soc. 2011, 133, 9104-9111
Nature, 2014, 509, 318-324.

The spiroindole protocol is beautifully exemplified with many examples that also show functional group tolerance. Many of the substrates are quite simple and as a result, these are a bit redundant. It would be more revealing if oxidation-sensitive substrates were included. Instead, the approach seems to be to pack in a lot of substrates for "shock and awe" value. It would be better if the scope were demonstrated in a more chemically interesting manner. But, this section is still good, and in line with today's practice.

The 2-oxindole syntheses with the simpler substrates will also be of interest to synthetic chemists. The control that is achieved in light of the overoxidation (Witkop pathway) is also very nice and likely of some utility. As this derives mostly from what appears to be a substituent effect and a solvent effect (Table 1 versus Table 3 versus Table 5), it would be helpful if the mechanistic study were conducted to yield conclusions and a more clear dichotomy along these lines. In this sense, it might make sense to show some of the same substrates, side by side under the different reaction conditions of each table. This would certainly make the paper easier to read.

But, this is superb chemistry, and likely to be of broad interest. Hopefully this review process will help the authors to present the science in a manner that is a bit easier to read and understand in terms of the nuances of the different reaction conditions and substrates. If so, Nature Communications seems like a great place for this impactful work.

Reviewer #2 (Remarks to the Author):

Tong et al. reported an efficient oxidation of indoles with oxone and KBr. Although the results are interesting, the main originality of the chemistry described in this manuscript has been reported in the literatures (Simon J. Garden, J. Braz. Chem. Soc., Vol. 30, No. 1, 19-32, 2019 and 15th Brazilian Meeting on Organic Synthesis – 15th BMOS – November 10-13, 2013 - Campos do Jordão, Brazil, the authors did not cite two papers). In addition, oxidation of organic compounds with oxone and KBr system has been well documented. Thus, this referee feels that this manuscript does not match the high standard of Nature Communications, therefore, I cannot recommend its publication by Nature Communications.

Reviewer #3 (Remarks to the Author):

In this manuscript, the authors reported an efficient halide catalysis for 3 major oxidations of

indoles using green conditions. The scope of the reactions is very broad. The methods and conditions are straight-forward. More importantly, a large variety of indole frameworks can be accessed quickly, most of which could be extended in natural product syntheses. Therefore, what the authors have presented is highly valuable for a broader synthetic community, and publication in this journal should be considered. I have the following points that need the authors to address in a revised version of the manuscript:

- Table 2, how does the electronic properties of the N-protecting group R1 and R3 influence the reactivity? Mechanistically?
- Table 2, explain why 0% for 2v? Why ester in 2u helped dr?
- Table 4, several products gave 0% yields, is this the weakness of the current method? How does it compare with other conventional methods? Point out the difference clearly.
- Table 4, substrate 7a and Table 6, substrate 12a or 12f, what will happen when you replace the methyl group with CF₃ group? Show some examples using trifluoromethylated indoles at C2/C3 positions. Many lit. examples have reported their synthesis yet not much is known about their applications in reactions such as the current manuscript.

NCOMMS-19-23569: Green Oxidation of Indoles using Halide Catalysis

Point-to-Point Response to Referees' comments/concerns:

Reviewer #1: Some of the uncited work is also relevant to the synthetically useful rearrangements noted in the present submission. So, these two citations really should be added.

Our Response:

We appreciate this referee on evaluating our manuscript and for her/his high recognition of our work. These two citations were added as ref 7 and ref 13 in revised our manuscript.

Reviewer #2: Tong et al. reported an efficient oxidation of indoles with oxone and KBr. Although the results are interesting, the main originality of the chemistry described in this manuscript has been reported in the literatures (Simon J. Garden, J. Braz. Chem. Soc., Vol. 30, No. 1, 19-32, 2019 and 15th Brazilian Meeting on Organic Synthesis – 15th BMOS – November 10-13, 2013 - Campos do Jordão, Brazil, the authors did not cite two papers). In addition, oxidation of organic compounds with oxone and KBr system has been well documented.

Our Response:

This referee might misunderstand our chemistry by comparing our work with the DMDO oxidation of indole (generally from oxone and acetone). Herein, we list some major differences and our novelty: (a) catalysis (oxone-halide) vs non-catalysis (oxone-acetone), oxone-halide involves halonium as an active catalyst while oxone-acetone generates DMDO as stoichiometric oxidant (not catalysis); therefore, our work is the first example of halide catalysis for green indole oxidation. (b) Different mechanism: oxone-halide involves halonium ion intermediate followed by water addition and Pinacol-type rearrangement, which is similar to NBS-mediated mechanism; DMDO effects the oxidative rearrangement through epoxidation pathway, which resembles m-CPBA-promoted mechanism. (c) Indole N-protecting group: DMDO requires electron-withdrawing group to protect the indole nitrogen while oxone-halide allows free indole (without N-protection) to be chemoselectively oxidized. As for the comment “oxidation of organic compounds with oxone and KBr system has been well documented”, we feel uncomfortable because the referee ignores two facts: 1) three

major types of indole oxidation are fundamentally important and 2) oxone-halide is used for the first time as a green catalytic system for these indole oxidations. To ease this referee's concern, we added the above mentioned paper as ref 14 in our revised version. Anyway, we respect different views on our chemistry.

Reviewer #3:

(1), "Table 2, how does the electronic properties of the N-protecting group R1 and R3 influence the reactivity? Mechanistically? "

Our Response:

The high chemoselectivity of indole oxidation via halide catalysis is hinged on that in situ generated halenium ion (c.f., Br⁺) as a catalyst reacts only with electron-rich indole (C2=C3) to form the corresponding indole halonium intermediate (**I**, Fig 2b). An electron-withdrawing group on the indole nitrogen (R1 = Ts, Boc) will substantially decrease the electron-density of indoles and consequently suppress the halenium-catalyzed indole oxidation, which is supported by the result of 2d with 0% yield (R1 = Ts or Boc) in Table 2. While electronic properties of N-protecting R3 is less significant to the electron density of indoles (not a conjugate system) and less influential to their oxidation under the halide catalysis. In fact, both electron-donating and withdrawing N-protecting groups (R3) are tolerated in our halide catalysis system (Table 2, [2i, 2j, 2w, 2x, 2y] vs [2e-2h]). The low yield for electron-donating R3 (2i, 2j, 2w, 2x, 2y) products is attributed to the difficult isolation/purification of tertiary amine products from our acidic reaction medium.

(2) "Table 2, explain why 0% for 2v? Why ester in 2u helped dr? "

Our Response:

Halenium-catalyzed oxidation of the C1-phenyl THC (1v) resulted in an unexpected oxidative cleavage of THCs to 2-phenylketo-N-Acetyltryptamine, which has been isolated with full characterization (see SI). We believe that the C1-phenyl (or C1-aromatic) group assists either halonium ion opening (**I**, Fig 2b) by elimination C1-proton or iminium-enamine tautomerization (**II**, Fig 2b), both of which would lead to C1-N3 cleavage after hydrolysis. This mechanistic explanation was included in the

Supplementary Information. As for impact of ester on dr, we could not have a good explanation, however, we did repeat the experiment to confirm the dr value was accurate. Apparently, it requires substantial experiments and computation work to better address its role on dr.

(3) “Table 4, several products gave 0% yields, is this the weakness of the current method? How does it compare with other conventional methods? Point out the difference clearly.”

Our Response:

The differences of our method from prior methods were summarized in the **Discussion** in the main text. One major weakness of our halide catalysis method is the requirement of electron-rich indole substrates, which exclude N-protecting electron-withdrawing groups on the indole nitrogen and electron-withdrawing substituent at C3 such as Phenyl or CF₃. In fact, indole (8a', 0%) is a poor substrate for many classical oxidations: *m*-CPBA gave a complex mixture while NBS afforded 40% yield with 60% conversion. N-Ts skatole (7d') resulted in a complex mixture under both NBS and *m*-CPBA.

(4) “Table 4, substrate 7a and Table 6, substrate 12a or 12f, what will happen when you replace the methyl group with CF₃ group? Show some examples using trifluoromethylated indoles at C2/C3 positions. Many lit. Examples have reported their synthesis yet not much is known about their applications in reactions such as the current manuscript.

Our Response:

As requested by this referee, we synthesized the C2/C3-CF₃ indole 7a''' and 12o and examined their oxidation under our halide catalysis system. The results were included in

the Table 4 and Table 6 and related discussion is also provided in the main text. We summarized these results here for your reference. CF₃ as a strong electron-withdrawing group at C2 or C3 lowers substantially the electron density of indoles and therefore disfavor the oxidation with halide catalysis. For example, 3-CF₃ indole (7a^{'''}) remains intact under our bromide catalysis for 20 hours while it decomposes under oxone-chloride-HFIP condition (Table 6, 13p). Intriguingly, 2-CF₃ skatole underwent Witkop oxidation and delivered 46% yield of 2-keto acetoniide 13o (Table 6).

REVIEWERS' COMMENTS:

Reviewer #3 (Remarks to the Author):

All of my comments have been addressed in the response letter and the revised manuscript, particularly glad to see the added entries of trifluoromethylated indoles. Now the manuscript is in a good shape to be accepted.